# Effects of Fe Impurities on Self-Discharge Performance of Carbon-Based Supercapacitors

**DOI:** 10.3390/ma14081908

**Published:** 2021-04-11

**Authors:** Yuting Du, Yan Mo, Yong Chen

**Affiliations:** 1State Key Laboratory of Marine Resource Utilization in South China Sea, Hainan Provincial Key Laboratory of Research on Utilization of Si-Zr-Ti Resources, Hainan University, Haikou 570228, China; 18080500210005@hainanu.edu.cn (Y.D.); moy0128@mail.nwpu.edu.cn (Y.M.); 2Guangdong Key Laboratory for Hydrogen Energy Technologies, School of Materials Science and Hydrogen Energy, Foshan University, Foshan 528000, China

**Keywords:** supercapacitors, self-discharge, carbon, redox reactions

## Abstract

Activated carbon is widely used as an electrode material in supercapacitors due to its superior electrochemical stability, excellent electrical conductivity, and environmental friendliness. In this study, the self-discharge mechanisms of activated carbon electrodes loaded with different contents of Fe impurities (Fe and Fe_3_O_4_) were analyzed by multi-stage fitting to explore the tunability of self-discharge. It is was found that a small quantity of Fe impurities on carbon materials improves the self-discharge performance dominated by redox reaction, by adjusting the surface state and pore structure of carbon materials. As the content of Fe impurities increases, the voltage loss of activated carbon with the Fe impurity concentrations of 1.12 wt.% (AF-1.12) decreases by 37.9% of the original, which is attributable to the reduce of ohmic leakage and diffusion, and the increase in Faradic redox at the electrode/electrolyte interface. In summary, self-discharge performance of carbon-based supercapacitors can be adjusted via the surface state and pour structure, which provides insights for the future design of energy storage.

## 1. Introduction

Supercapacitors (SCs) are promising power storage devices due to their high-power density, ultra-long cycle life, reliable safety, and environmental friendliness [1,2,3]. In recent decades, the preparation process of novel SCs with high energy density and a large potential window has developed considerably. The supercritical drying process has been considered to be a promising method for the electrode materials of novel SCs. For example, an aerogel containing dispersed graphene as the porous electrode of novel SCs was fabricated using a supercritical CO_2_ process, which exhibited a high energy density of 79.2 Wh kg^−1^ at a power density of 0.23 KW kg^−1^ [4]. Furthermore, porous carbon microspheres as the electrode materials of SCs were prepared in a closed autoclave under high pressure, and exhibited a high capacitance of 633.2 mA h g^−1^ at 50 mA g^−1^ [5]. A carbon textile electrode was synthesized by vacuum impregnation and spray coating, resulting in excellent cycle performance at 1.5 V [6]. However, the application of novel SCs is limited by a rapid self-discharge (SDC) rate. SDC can be defined as a thermodynamic spontaneous process of energy decay, the appearance of which leads to low energy retention and large voltage loss [7,8]. The electrode material is an important factor that determines the SDC performance of SCs [9,10,11], due to its complex surface structure and chemistry state [2,12,13,14]. Activated carbon has attracted increasing attention in electrode materials due to its high specific surface area, adjustable pore structure, superior electrochemical performance, and low cost [15,16,17]. Thus, the SDC mechanisms of activated carbon have been further studied to inhibit voltage loss for electrochemical performance.

To date, numerous studies have aimed to improve the SDC of carbon material by adjusting its surface structure and functional groups [14,18,19,20]. For instance, Wang et al. constructed nano-layers on the carbon surface, reducing the 50% SDC rate of the original [13]. Davis and coworkers distinguished the effects between carbon oxidation and diffusion on SDC, and proposed that the redox reaction (17% of SDC) could be the main factor of SDC for activated carbon [14]. Moreover, Yuan et al. proposed that removing the oxygen functional group of carbon by hydrogen thermal reduction can reduce the voltage loss of SDC, due to the increased stability of reaction at the electrode/electrolyte interface [1]. However, the self-discharge mechanism of Fe and its oxides on the carbon surface has not yet been identified in KOH electrolyte [2,12,21,22,23,24]. Therefore, in-depth study of the SDC mechanism caused by Fe impurities, in addition to clarifying the effects of their contents on SDC of carbon materials, are of significant importance for the development of advanced SC devices [25,26,27].

In this study, different contents of Fe impurities loaded on purified activated carbon (Fe@C composites) were synthesized by the vacuum impregnation process followed by annealing at 400 °C for 2 h in an effort to establish a correlation between voltage loss during the SDC process, and contents of Fe and its oxides in SCs. The influences of Fe impurities on the structures and properties of the resulting samples were studied by scanning electron microscopy (SEM) and transmission electron microscopy (TEM). The crystal structures and valence states of Fe@C composites were investigated by X-ray diffraction (XRD), Raman spectroscopy, and X-ray photoelectron spectroscopy (XPS). The correlation between SDC performance and contents of Fe impurities was systematically investigated using MATLAB software. The results show that voltage loss during the SDC process could be effectively suppressed by introducing low-content Fe impurities in carbon electrodes.

## 2. Experimental

All reagents were purchased from Aladdin and used as received without further purification. Commercial activated carbon (AC, Hainan Xingguang Activated Carbon Co., Ltd., Wenchang, China) powder was washed by 10 vol.% HCl and sufficient deionized water to remove impurities, followed by drying at 80 °C for 12 h. Fe@C was prepared by first dissolving FeCl_3_·9H_2_O in distilled water to form a transparent solution. Next, AC (8 g) was added into the above dispersion containing different concentrations of FeCl_3_ solution (0.48, 0.80, 1.12, and 1.6 wt.%) followed by vacuum impregnation and ultrasonic dispersion for 1 h. The resulting homogeneously dispersed mixtures were filtered and dried at 120 °C for 8 h. Finally, the obtained powders were annealed at 400 °C for 2 h under argon flow to yield the samples AF-0.48, AF-0.8, AF-1.12, and AF-1.6.

The morphological microstructures of the samples were studied by scanning electron microscopy (SEM, Hitachi S-4800SEM, Hitachi, Tokyo, Japan), transmission electron microscopy (TEM, JEM-2100F, JEOL Ltd., Tokyo, Japan) and high-resolution transmission electron microscopy (HRTEM, JEM-2100F, JEOL Ltd., Tokyo, Japan). The crystal structures were obtained by X-ray diffraction (XRD, Bruker D8 advance, Bruker, Karlsruhe, Germany) with Cu-Kα radiation in the scanning range from 10° to 80° and Raman spectroscopy (Thermo Fisher DXRxi, Waltham, MA, USA) under a laser excitation wavelength of 532 nm. The energy dispersive spectroscopy (EDS, Hitachi, Tokyo, Japan) and X-ray photoelectron spectroscopy (XPS, ESCALAB 250xi, Thermo Fisher Scientific, Waltham, MA, USA) profiles were collected and used to analyze the surface chemical compositions of samples. The amounts of Fe in AC and Fe@C composites were collected by inductively coupled plasma mass spectrometry (ICP-MS, Agilent ICP-MS-7700, Agilent Technologies Inc., Santa Clara, CA, USA). The specific surface areas of the samples were collected using the Brunauer–Emmett–Teller (BET) method on an AutoSorb iQ2 under liquid argon at 87 K. The pore size distribution was calculated using the quenched-solid density functional theory (QSDFT) model based on the adsorption branch.

The Fe@C composite electrodes used for the fabrication of SCs were prepared by mixing active materials (load Fe/Fe_3_O_4_ carbon powder, 85 wt.%), conductive agent (Ketjen black, 10 wt.%), and binder (polytetrafluoroethylene, 5 wt.%) in ethyl alcohol to form a slurry. After mixing and shaking, the slurry was pressed on Ni foam (1 × 1 cm^2^) and left to dry at 110 °C under vacuum for 12 h. The loading mass of the slurry on the electrodes was about 2 mg cm^−2^. Next, two Fe@C composite electrodes with the same mass were assembled into symmetrical SCs, in which 6 M KOH was employed as an electrolyte. The cycle performance and galvanostatic charge/discharge (GCD) profiles were tested using a Neware battery testing system at the charging current density of 1 A g^−1^ and 25 °C. The cyclic voltammetry (CV) curves were collected by Bio-logic VSPN-300 in the potential window from 0 to 1 V at the scan rate of 1 mV s^−1^. The electrochemical impedance spectroscopy (EIS) curves were collected in the frequency range from 1 mHz to 100 kHz at 5 mV amplitude. The 24 h SDC curves were measured using a Neware battery testing system, and used to study the influence of iron impurity on the SDC process of carbon-based SCs.

## 3. Results and Discussion

The surface morphologies of Fe@C composites were investigated by SEM and the results are displayed in Figure 1a. The annealed Fe@C composites present irregular particles with porous structure. The porosity between the disordered particles and porous structure provides a favorable space for the penetration and diffusion of the electrolyte, which is useful for improving the capacitance of the SCs [28,29,30]. In addition, the Fe, C, and O elements are also uniformly distributed on the particle surface (Figure 1b). The crystal structures of Fe@C composites were further investigated by TEM and HRTEM measurements. As shown in Figure 1c,d, Fe and its oxides are uniformly anchored on the activated carbon surface, and the average size of particles in AF-1.12 is about 4.5 nm (inset picture of Figure 1c). Furthermore, the porous structure of activated carbon particles provides enough space for the multiphase changes of Fe_3_O_4_ to prevent significant volumetric expansion during the charge/discharge processes to maintain high cycling stability [31,32,33,34]. More direct evidence regarding the structure of Fe@C composites is illustrated in Figure 1d. For the AF-1.12 sample, the lattice distances are estimated to be 0.253 nm and 0.202 nm and attributed to (331) lattice planes of Fe_3_O_4_ and (110) lattice planes of Fe, respectively [35,36,37,38,39]. Thus, Fe/Fe_3_O_4_ was successfully loaded on carbon using an impregnation process followed by annealing.

XRD and Raman spectroscopy were employed to further investigate the crystallinity and structure of Fe@C composites. The characteristic diffraction peaks at 2θ = 35.42° and 63° correspond to the reflection planes of (311) and (440) in Fe_3_O_4_ with a face-centered cubic structure [36,37,38,40,41,42]. The diffraction peak at 2θ = 44.73° is assigned to the (110) lattice planes of body-centered-cubic Fe [36,40,43]. As Fe-doped content increases, the peak intensity of Fe_3_O_4_ and Fe increases, implying enhanced Fe impurities on the Fe@C composite surface. In addition, the Fe contents of Fe@C composites were collected by ICP-MS, and are larger than that of AC (425 mg kg^−1^) (Appendix A). As the Fe-doping content increases, the Fe contents of AF-0.48, AF-0.8, AF-1.12, and AF-1.6 increase to 3260.2, 4625.3, 6952.0, and 9911.5 mg kg^−1^, respectively. To further study the effect of Fe-doping on surface structure of active carbon, Raman spectroscopy was carried out and the results are shown in Figure 2b. All samples exhibit two typical carbon peaks at 1360 cm^−1^ (D-band) and 1580 cm^−1^ (G-band) [42,44,45], ascribed to the amorphous and graphitic carbon structure [46,47], respectively. The intensity ratio of the D-band and G-band (I_D_/I_G_) reflects the degree of surface disorder of carbon materials. The calculated I_D_/I_G_ values of the samples AC, AF-0.48, AF-0.8, AF-1.12, and AF-1.6 are 1.05, 1.06, 1.07, 1.08, and 1.10, respectively. This result indicates that the degree of surface disorder is more pronounced, which is attributed to the increase in Fe impurities [37,48,49]. In addition to SEM and TEM analyses, the XRD patterns and Raman spectra also confirm the formation of Fe_3_O_4_ and Fe phases in activated carbon through the impregnation process.

To determine the valence states of Fe and its oxides, XPS measurements were carried out. As shown in Appendix A, the XPS survey scan unambiguously indicates the presence of C, O, and Fe elements. The increase in Fe/Fe_3_O_4_ content leads to a gradual enhancement in the peak intensity of Fe 2p corresponding to the Fe@C composite when compared to that of AC. The high-resolution spectra of C 1s are divided into six peaks at of 284.7, 285.4, 286.2, 287.4, 288.7, and 290 eV, corresponding to sp^2^ (graphitic carbon, C-C peak), sp^3^ hybridized carbon (defects peak), C–O, C=O, O–C=O, and π–π*, respectively (Figure 2c) [24,43,50,51]. As shown in Appendix A and Appendix A, the increase in Fe-doping content leads to the gradual transformation from sp^2^-C to sp^3^-C, indicating the decrease in graphitization degree [51]. This result is consistent with the XRD and Raman results. In Figure 2d, the O 1s could be deconvoluted into three peaks at 530.2, 531.8, and 533.2 eV, corresponding to O–Fe of Fe_3_O_4,_ O=C, and O–C, respectively [40,41,43,50]. In the Fe 2p spectrum of AF-1.12 (Figure 2e), five peaks at 710.8, 712.7, 719.0, 724.5, and 725.2 eV are observed. These are associated with, respectively, Fe 2p_3/2_, shake-up satellite and Fe 2p_1/2_ of Fe^2+^ and Fe^3+^, indicating the presence of Fe_3_O_4_ [28,36,37,50]. As a result, a unique complex system consisting of Fe@C with different doping concentrations was obtained. 

The surface areas and pore size distributions of AC and Fe@C composites were investigated by Ar adsorption isotherms at 87 K. As shown in Figure 3a, all samples display a combined Ι/IV isotherm [15,35,37,42,49,52,53]. Specifically, the rapid increase in isotherms in the low-pressure region is a typical characteristic of a micro-porous structure [54]. The adsorption/desorption isotherms in the high-pressure region exhibit a non-obvious platform and the type H4 hysteresis loops, indicating the presence of meso-pores [35]. The effects of Fe/Fe_3_O_4_ on the surface pore size distributions were further studied using the QSDFT method. As shown in Figure 3b, all samples have micro-pores (pore size < 2 nm) and meso-pores (pore size between 2 and 50 nm), and the volume of the former is higher than that of latter. In Appendix A, as Fe-doping content rises, the specific area increases from 1503.9 m^2^ g^−1^ (AC) to 1676.81 m^2^ g^−1^ (AF-1.12) and then decreases to 1586.46 m^2^ g^−1^ (AF-1.6). AF-1.2 exhibits the highest specific area of 1671.81 m^2^ g^−1^, 11.2% higher than that of AC. The larger surface area can be attributed to the increase in the microporous volume by the etching effect of Fe_3_O_4_ during the heat treatment process [55,56] (Figure 3b). The specific area of AF-1.6 is 1586.46 m^2^ g^−1^, which is smaller than that of AC-1.2. The reduction of specific surface area can be attributed to the reduction of the microporous volume, which is due to the destruction of the microporous structure on the carbon surface caused by the massive growth of Fe/Fe_3_O_4_ [57].

The capacitive performances of AC and Fe@C composites were investigated in 6 M KOH for over 10,000 cycles. As shown in Figure 4a, as Fe-doping content rises, the specific capacitance increases from 146.7 F g^−1^ (AC) to 186.2 F g^−1^ (AF-1.12) and then decreases to 173.2 F g^−1^ (AF-1.6). AF-1.2 exhibits the highest specific capacitance of 186.2 F g^−1^, 26.9% higher than that of AC. The enhanced electrochemical performance can be attributed to the introduction of Fe/Fe_3_O_4_, which increases the pseudo-capacitance by redox reactions in the interface between the electrode and the electrolyte. The specific capacitance of AF-1.6 is 173.2 F g^−1^, which is smaller than that of AC-1.2. The reduction of specific capacitance can be attributed to the reduction of the double-layer capacitance caused by the destruction of the microporous structure and the reduction of specific surface areas. After 10,000 cycles, the specific capacitance of AF-1.12 remained at 178 F g^−1^, indicating excellent cycle stability (Figure 4a). The degradation in long-cycle performance is caused by the instability of increased pseudo-capacitance by Fe and its oxides. The GCD curves of Fe@C composites appear nonlinear and asymmetrical, suggesting a strong pseudo-capacitive behavior (Figure 4b) [44,51,57].

To gain further insights into the effects of Fe and Fe_3_O_4_ on the electrochemical performances, CV and EIS curves were recorded and the results are gathered in Figure 4c,d. The CV curves of Fe@C composites display asymmetrical pseudo-capacitance behavior (Figure 4c), corroborating the GCD curves. The asymmetrical CV curve of AF-1.12 suggests the presence of pseudo-capacitance during the charge and discharge processes [41,58]. In Figure 4d, the X-intercept of the semicircles in the Nyquist plots in the high-frequency region is related to the ohmic resistance of the SCs (R_ohm_), which represents the total resistance produced by the electrolyte, current collectors, and electrodes [28,29]. Among all of the samples, the AC exhibits the highest R_ohm_ of 3.88 Ω, which can be attributed to the low charge-transfer rate caused by its uneven surface and the intricate conductive network. With increases in the loading amounts of Fe and Fe_3_O_4_, the R_ohm_ values decrease to 2.57, 1.08, 0.94, and 0.71 Ω for AF-0.48, AF-0.8, AF-1.12, and AF-1.6, respectively. Such a significant decrease occurred because the Fe element contributed to structural rearrangement and more active sites on the carbon surface, which accelerates charge separation and transport [59,60,61,62]. In the low-frequency region, the impedance plots of all samples become oblique lines, showing pure capacitance.

The main form of self-discharge of charged SCs relies on voltage attenuation [12,63]. The devices undergo two and a half cycles of charge/discharge (at 1 A g^−1^) and then hold at 1 V for 2 h to reduce the influence of uneven charges on the electrode surface. As shown in Figure 5a, the potential of AC drops to 0.27 V after 24 h, equivalent to 73% voltage loss in the SCs. By comparison, AF-0.48 takes the same time to drop its potential to 0.46 V, equivalent to only 54% voltage loss. The increase in Fe and Fe_3_O_4_ contents leads to the gradual decline in the voltage loss of SDC. Compared with the potential decay of AC, that of AF-1.12 is reduced by 39.7% (0.56 V, equivalent to only 44% voltage loss) after 24 h. According to the mathematical models proposed by Conway et al., the potential decay is caused by three mechanisms of SCs, consisting of ohmic leakage, Faraday reaction, and diffusion (Figure 6) [12,14]. The potential decay is caused by the internal resistances between the positive and negative electrodes, which can be described by the ohmic leakage mechanism in Figure 6b (V ~ exp(t)) [63,64]. As shown in Figure 6c, the potential decay is attributed to the departure of ions from the electric double layer during the SDC process, which is the diffusion mechanism (V ~ t^1/2^) [12,24]. The potential decay is related to the redox reactions of impurities on the electrode surface, which can be described by the Faradic redox mechanism (V ~ ln (t)) (Figure 6d) [14,24,62]. Combining the potential decay caused by all three mechanisms, Equation (1) can be obtained [63,65].
(1)$$U=U_{0}-\mathrm{Aexp}\left(t\right)-{\mathrm{Bt}}^{\frac{1}{2}}-\mathrm{Dln}\left(t\right)-e$$
where U and U_0_ represent the final potential and initial potential (1.0 V) of the charged SC. t is the discharge time (24 h). A and B correspond to the time constant and diffusion parameter of the SDC process, respectively. e and D denote the constants in the Faradaic process, related to the current density.

The effects of Fe/Fe_3_O_4_ on SDC were further studied by establishing the SDC mechanisms by multi-stage fitting in MATLAB. In Figure 5a–c, the SDC process consists of three parts. The ohmic leakage mechanism represents the first part (red curve), the Faradic redox mechanism displays the second part (green curve), and the diffusion mechanism shows the last part (blue curve). The fitting curves match well with the test results (black curve), confirming the complex nature of the SDC process from the ohmic leakage mechanism to the diffusion mechanism. V_d_ represents the occurrence of the voltage of the diffusion mechanism. As shown in Figure 5d, as the content of Fe-doping increases, the decrease in R_ohm_ leads to a decline in SDC controlled by the ohmic leakage mechanism. In addition, with an increase in Fe-doping content, the reduction of micro-pores leads to a decrease in V_d_, indicating the weakening of SDC caused by diffusion mechanism. By contrast, the redox of Fe and Fe_3_O_4_ enhances the SDC controlled by the Faradic redox mechanism, corroborating the CV and GCD data. However, SDC can increase again at high doping levels, which is due to the destruction of the surface structure by volume expansion of Fe_3_O_4_ (Appendix A). As a result, the redox reactions can change the voltage loss of SDC at low contents of Fe and its oxides.

## 4. Conclusions

Different contents of Fe/Fe_3_O_4_ were successfully introduced into activated carbon by vacuum impregnation followed by annealing. The influence of Fe-doping content on SDC of carbon materials was systematically investigated. The proportion of ohmic leakage, redox, and diffusion mechanism of SDC processes was analyzed using MATLAB. The introduction of Fe/Fe_3_O_4_ at low contents into activated carbon (AF-1.12) leads to a drastic reduction in SDC, of more than 39.7%, which was attributed to the decreased decay of potential caused by the diffusion mechanism and ohmic leakage mechanism in charged SCs. AF-1.12 showed the highest specific capacitance (186.2 F g^−1^) and excellent cycle stability, thus showing promise for the improvement of small power storage devices. 

## Figures and Tables

**Figure 1 materials-14-01908-f001:**
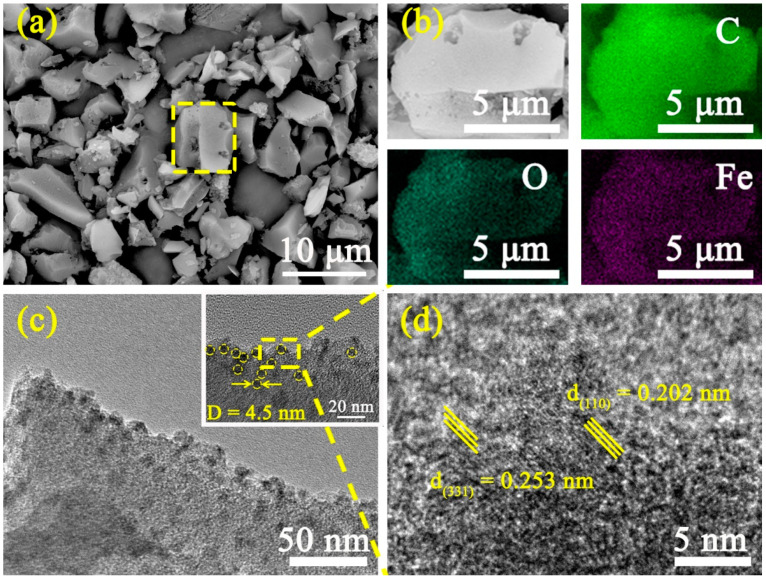
(**a**) SEM images; (**b**) EDS results; (**c**) TEM, and (**d**) HRTEM images of AF-1.12 sample.

**Figure 2 materials-14-01908-f002:**
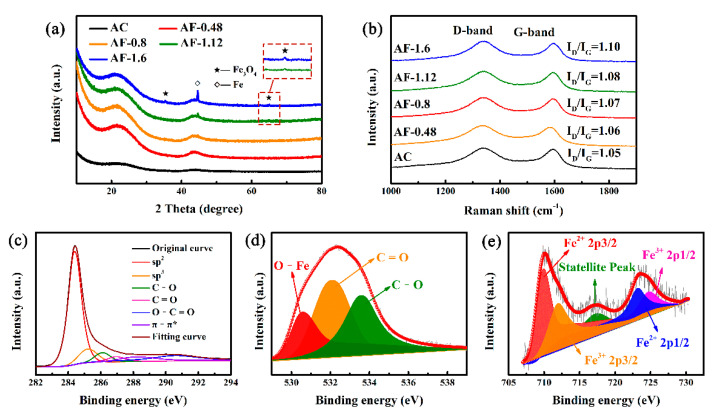
(**a**) XRD patterns and (**b**) Raman spectra of AC and Fe@C composites; high resolution XPS survey of AF-1.12: (**c**) C 1s, (**d**) O 1s, and (**e**) Fe 2p.

**Figure 3 materials-14-01908-f003:**
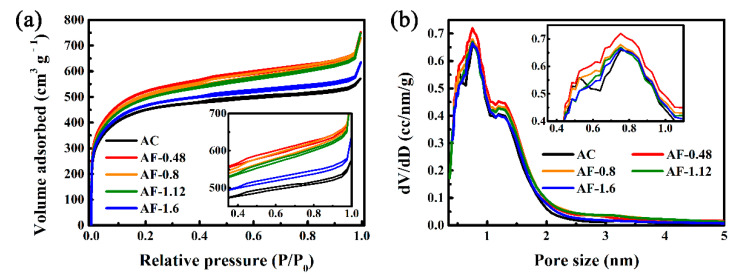
Ar adsorption/desorption isotherms (**a**) and corresponding pore size distributions (**b**) of AC and Fe@C composites.

**Figure 4 materials-14-01908-f004:**
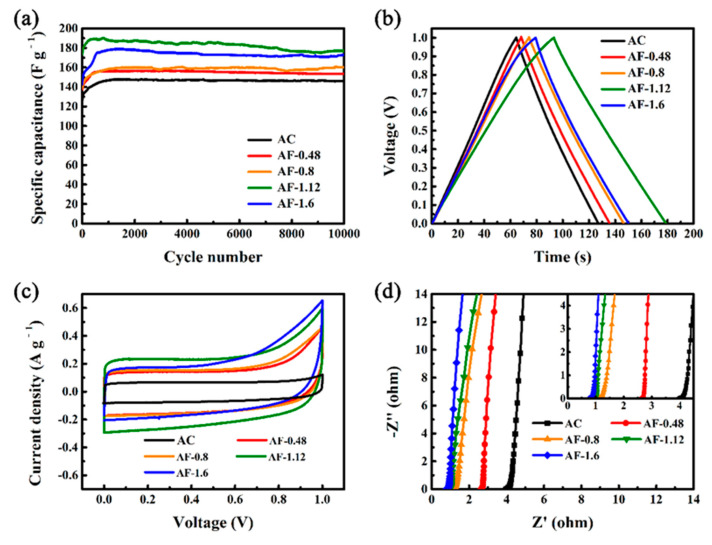
Cycle performances (**a**); galvanostatic charge/discharge (GCD) curves (**b**) at 1 A g^−1^; CV curves (**c**) at 1 mV/s, and Nyquist plots (**d**) of AC and Fe@C composite supercapacitor (SC) electrodes.

**Figure 5 materials-14-01908-f005:**
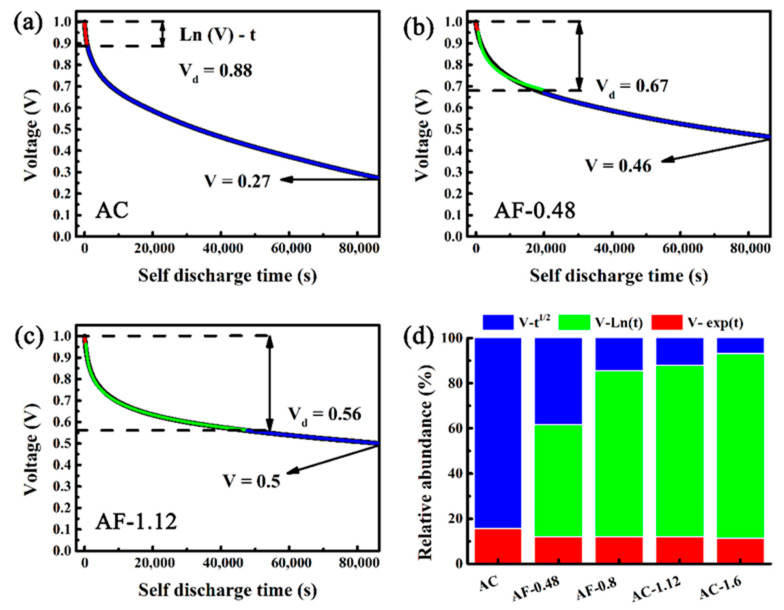
Simulated self-discharge (SDC) of AC and Fe@C composite electrodes by multiple processes: (**a**) AC electrode, (**b**) AF-0.48 electrode, and (**c**) AF-1.12 electrode. The control time of different self-discharging processes (**d**).

**Figure 6 materials-14-01908-f006:**
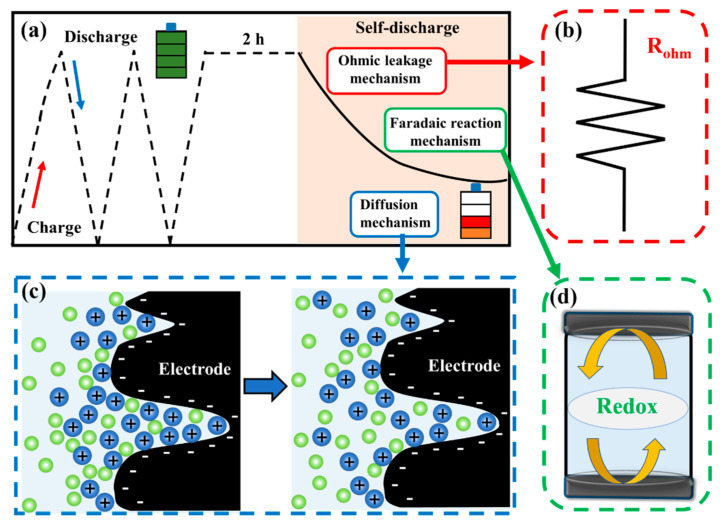
(**a**) Illustration of the experiment; The mechanisms of SDC in SCs: (**b**) ohmic leakage mechanism; (**c**) diffusion mechanism; and (**d**) Faraday reaction mechanism.

## Data Availability

All data is contained within the article.

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
