# Peer review of "Effects of Fe Impurities on Self-Discharge Performance of Carbon-Based Supercapacitors"

_materials, 2021, doi:10.3390/ma14081908_

Round 1
Reviewer 1 Report
The manuscript “Effects of Fe Impurities on Self-discharge Performance of Carbon-based Supercapacitors” deals with the analysis of the self-discharge mechanisms of activated carbon electrodes loaded with different contents of Fe impurities, using a multi-stage fitting. Several analyses have been performed, obtaining good results. Therefore, the publication of this study is recommended; but after some revisions.
In particular:
- Introduction. The state of the art about the preparation of novel supercapacitors has to be enlarged, to highlight the novelty of this work. For this purpose, see, for example, the work of Sarno et al., SC-CO2-assisted process for a high energy density aerogel supercapacitor: The effect of GO loading, Nanotechnology, 2017, 28, Article number 204001; etc..
- Results and discussion. Please, explain the influence of the morphology and porosity on the mechanical and electrochemical performance of the samples produced.
Author Response
Response to Reviewer 1 Comments
Thanks for reviewers’ comments concerning our manuscript entitled “Effects of Fe Impurities on Self-discharge Performance of Carbon-based Supercapacitors” (Manuscript ID: materials-1163098). These comments are all valuable and very helpful for revising and improving our paper, as well as the important guiding significance to our researches. We have studied comments carefully and have made corrections which we hope meet with approval.
Reviewer 1: The manuscript “Effects of Fe Impurities on Self-discharge Performance of Carbon-based Supercapacitors” deals with the analysis of the self-discharge mechanisms of activated carbon electrodes loaded with different contents of Fe impurities, using a multi-stage fitting. Several analyses have been performed, obtaining good results. Therefore, the publication of this study is recommended; but after some revisions.
In particular:
Point 1: Introduction. The state of the art about the preparation of novel supercapacitors has to be enlarged, to highlight the novelty of this work. For this purpose, see, for example, the work of Sarno et al., SC-CO2-assisted process for a high energy density aerogel supercapacitor: The effect of GO loading, Nanotechnology, 2017, 28, Article number 204001; etc.

Response 1: Thanks for review’s kind reminder. As reviewer suggested, we have added the related section and new published to the revised manuscript (Page 1, line 2) as follow:
- Introduction
Supercapacitors (SCs) are promising power storage devices owing to their high-power density, ultra-long cycle life, good safety, and environmental friendliness [1-3]. In the past decades, the preparation process of novel supercapacitors with high energy density and large potential window has developed considerably. Among them, the supercritical drying process has been considered as a promising preparation process of novel supercapacitor. For examples, a compact and porous Poly(vinylidene fluoride-co-hexafluoropropylene) containing dispersed graphene was fabricated by a supercritical CO2 assisted gel drying process, which exhibited high energy density of 79.2 Wh kg-1 at a power density of 0.23 KW kg-1 [4]. Also, porous carbon microspheres as the electrodes of supercapacitor was fabricated in the closed autoclave under high pressure, which exhibited high capacitance of 633.2 mA h g-1 at 50 mA g-1 [5]. A solid-state supercapacitor was synthesized by vacuum impregnation and spray coating, capacitance of which reduced 1% of the original after 12,500 cycles [6]. However, the application in large energy storage equipment is limited by a rapid self-discharge (SDC) rate of SCs. SDC can be defined as a thermodynamic spontaneous process of energy decay, the appearance of which leads to low energy retention and large voltage loss [7, 8]. The electrode materials are important factors determining the SDC performance of SCs [9-11], which is due to their complex surface structure and chemistry state [2, 12-14]. Among materials, activated carbon has attracted increasing attention due to its high specific surface area, adjustable pore structure, superior electrochemical performance, and low-cost [15-17]. Thus, the SDC mechanism of activated carbon have been further studied to inhibit voltage loss for electrochemical performance.
References:
[4] Sarno, M.; Baldino, L.; Scudieri, C.; Cardea, S.; Reverchon, E. A one-step SC-CO2 assisted technique to produce compact PVDF-HFP MoS2 supercapacitor device. Journal of Physics and Chemistry of Solids 2020, 136, 109132.
[5] Qian, Y.; Jiang, S.; Li, Y.; Yi, Z.; Qian, Y. Understanding mesopore volume enhanced extra capacity: optimizing mesoporous carbon for high rate and long-life potassium storage. Energy Storage Materials 2020, 29, 341-349.
[6] Yong, S.; Owen, J.; Beeby, S. Solid state supercapacitor fabricated in a single woven textile layer for e-textiles applications. Advanced Engineering Materials 2018, 20, (5), 1700860.
Point 2: Results and discussion. Please, explain the influence of the morphology and porosity on the mechanical and electrochemical performance of the samples produced.
Response 2: As reviewer suggested, we have presented a more detailed statistical analysis on the electrochemical section (Page 6, line 2).
The capacitive performances of AC and Fe@C composites are investigated in 6 M KOH for over 10,000 cycles. As shown in Fig. 4a, as Fe-doping content rises, the specific capacitance increases from 146.7 F g−1 (AC) to 186.2 F g−1 (AF-1.12) and then decreases to 173.2 F g−1 (AF-1.6). AF-1.2 exhibits the highest specific capacitance with 186.2 F g-1, 26.9 % higher than that of AC. The enhanced electrochemical performance can be attributed to the introduction of Fe/Fe3O4, which increases the pseudocapacitance by redox reactions in the interface between electrode and electrolyte. The specific capacitance of AF-1.6 is 173.2 F g−1, which is smaller than that of AC-1.2. The reduce of specific capacitance can be attributed to the reduction of a partial of double-layer capacitance caused by the destruction of microporous structure and the reduce of specific surface areas. After 10,000 cycles, the specific capacitance of AF-1.12 remains at 178 F g−1, indicating excellent cycle stability (Figure 4a). The degradation in long-cycle performance is caused by the instability of increased pseudocapacitance by Fe and its oxide. The GCD curves of Fe@C composites look nonlinear and asymmetrical, suggesting a strong pseudocapacitive behavior (Fig 4b) [44, 51, 58].
Figure 4. Cycle performances (a), GCD curves (b) at 1 A g−1, CV curves (c) at 1 mV/s, and Nyquist plots (d) of AC and Fe@C composites SCs electrodes.

Reviewer 2 Report
The manuscript deals with the preparation of Fe doped activated carbon for supercapacitor electrode. The resultant AF electrodes showed improved self-discharge performance. And, the authors tried to correlate the doping amount with electrochemical performance in various ways. However, there are some points should be corrected and reconsidered. Thus, this manuscript should be modified before consideration of publishing in this journal.
(1) The amounts of Fe in AF samples are nominal wt%. Experimental results such as TGA or ICP are required.
(2) Figure 1; What is the average size of the Fe or Fe3O4 particles in AF sample?
(3) Figure 3a; The authors claimed that all samples exhibited type â…£ isotherms. However, the hysteresis is not clearly observed in Figure 3a.
(4) Table S2; The specific surface area of AF samples decreased with increasing Fe contents due to destruction of microporous structure. However, the bare AC showed a smaller specific surface area than that of AF-0.48. What is the reason for the discrepancy?
(5) Figure 4d; The authors claimed that the Rohm values decreased due to the growth of Fe content in the nanocomposites with high electrical conductivity. However, Fe/Fe3O4 composite might have much lower electrical conductivity compared to the carbon. Explain this.
Author Response
Response to Reviewer 2 Comments
Thanks for reviewers’ comments concerning our manuscript entitled “Effects of Fe Impurities on Self-discharge Performance of Carbon-based Supercapacitors” (Manuscript ID: materials-1163098). These comments are all valuable and very helpful for revising and improving our paper, as well as the important guiding significance to our researches. We have studied comments carefully and have made corrections which we hope meet with approval.
Reviewer 2:
The manuscript deals with the preparation of Fe doped activated carbon for supercapacitor electrode. The resultant AF electrodes showed improved self-discharge performance. And, the authors tried to correlate the doping amount with electrochemical performance in various ways. However, there are some points should be corrected and reconsidered. Thus, this manuscript should be modified before consideration of publishing in this journal.
Point 1: The amounts of Fe in AF samples are nominal wt%. Experimental results such as TGA or ICP are required.
Response 1: Thanks for review’s kind reminder. We have added the related ICP results in the revised manuscript (Page2, line 44, Page 4, line 5 and Page 9, line 3) and supporting information (Table. S1).
Experimental:
The morphological microstructures of the samples were studied by scanning electron microscopy (SEM, Hitachi S-4800SEM) and transmission electron microscopy (TEM, JEM-2100F, Japan). The crystal structures were obtained by X-ray diffraction (XRD, Bruker D8 advance) with Cu-Kα radiation in the scanning range from 10° to 80° and Raman (Thermo Fisher DXRxi, America) under a laser excitation wavelength of 532 nm. The energy dispersive spectroscopy (EDS) and X-ray photoelectron spectroscopy (XPS) profiles were collected by SEM (Hitachi S-4800SEM and XPS, Thermo Escalab250), and used to analyze the surface chemical compositions of samples. The amounts of Fe in AC and Fe@C composites were collected by inductively coupled plasma mass spectrometry (ICP-MS, Agilent ICP-MS-7700). The specific surface areas of the samples were collected by the Brunauer-Emmett-Teller (BET) method on an AutoSorb iQ2 under liquid argon at 87 K. The pore size distribution was calculated by quenched-solid density functional theory (QSDFT) model based on the adsorption branch.
Results and discussion:
The XRD and Raman spectroscopy are employed to further investigate the crystallinity and structure of each Fe@C composite. The characteristic diffraction peaks at 2 θ = 35.42° and 63° correspond to the reflection planes of (311) and (440) in Fe3O4 with face-centered cubic structure [36-38, 40-42]. The diffraction peak at 2 θ = 44.73° is assigned to the (110) lattice planes of body-centered-cubic Fe [36, 40, 43]. As Fe-doped content increases, the peak intensity of Fe3O4 and Fe rise, implying enhanced Fe impurities on the Fe@C composite surface. Also, the Fe contents of Fe@C composites are collected by ICP-MS, which are larger than AC (425 mg kg-1) (Table. S1). As the Fe-doping content increases, the Fe contents of AF-0.48, AF-0.8, AF-1.12, AF-1.6 increase to 3260.2, 4625.3, 6952.0, 9911.5 mg kg-1, respectively. To further verify the increasing Fe-doping in Fe@C composites, Raman spectroscopy is carried out and the results are gathered in Fig. 2b. All samples exhibit two typical carbon peaks at 1360 cm-1 (D-band) and 1580 cm-1 (G-band) [42, 44, 45], ascribed to the amorphous and graphitic carbon structure [46, 47], respectively. The intensity ratio of the D-band and G-band (ID/IG) would reflect the degree of surface disorder and graphitization of carbon materials. The calculated ID/IG values of the samples AC, AF-0.48, AF-0.8, AF-1.12 and AF-1.6 are 1.05, 1.06, 1.07, 1.08 and 1.10, respectively, indicating the structural disordering of Fe@C composite is more pronounced [37, 48, 49]. In addition to SEM and TEM analyses, the XRD patterns and Raman spectra also confirm the formation of Fe3O4 and Fe phases in activated carbon through the impregnation process.
Table S1. The amounts of Fe in AC and Fe@C composites are investigated by ICP-MS.
AC |
AF-0.48 |
AF-0.8 |
AF-1.12 |
AF-1.6 |
|
Fe (mg kg-1) |
425.0 |
3260.2 |
4625.3 |
6952.0 |
9911.5 |
Supplementary Materials: Fig S1: X-ray photoelectron wide scan spectra (a) and Fe 2p spectra (b) of AC and the Fe@C composites; Fig S2: C1s spectra of AC (a), AF-0.48 (b), AF-0.8 (c) and AF-1.6 (d); Table S1. The amounts of Fe in AC and Fe@C composites are investigated by ICP-MS. Table S2: Atomic percentage concentrations (%) of the peak area at C, O and Fe binding energy in the XPS spectra (a) and Percentage (%) of the peak area at sp2 and sp3 binding energy in the XPS spectra of C1s; Table S2: Specific surface areas of AC and Fe/Fe3O4 composites are investigated by Ar adsorption isotherms at 87 K; Fig S3: Self-discharge curves of AF-0.8 and AF-1.6.
Point 2: What is the average size of the Fe or Fe3O4 particles in AF sample?
Response 2: We have given an enlarged view of the morphology of Fe or Fe3O4 particles. It can be seen from the inset picture of Figure1c, the average size of Fe or Fe3O4 particles in AF-1.12 is about 0.45 nm. Also, we have revised the SEM section and Figure 1c as follow (Page 3, line 20):
The surface morphologies of Fe@C composites are investigated by SEM and the results are displayed in Fig. 1a. The annealed Fe@C composites present irregular particles with porous structure. The porosity between the disordered particles and porous structure would provide favorable space for the penetration and diffusion of the electrolyte, useful for improving the capacitance of the SCs [28-30]. Besides, the Fe, C, and O elements are also uniformly distributed on the particle surface (Fig. 1b). The crystal structures of Fe@C composites are further investigated by TEM and HRTEM measurements. As shown in Fig. 1c and 1d, Fe and its oxide are uniformly anchored on the activated carbon surface, and the average size of particles in AF-1.12 is about 0.45 nm (inset picture of Figure 1c). Meanwhile, the porous structure of activated carbon particles could provide enough space for the multiphase changes of Fe3O4 and prevent huge volumetric expansion during the charge-discharge processes to maintain high cycling stability [31-34]. More direct evidence regarding the structure of Fe@C composites is illustrated in Fig. 1d. For AF-1.12 sample, the lattice distances are estimated to 0.253 nm and 0.202 nm attributed to (331) lattice planes of Fe3O4 and (110) lattice planes of Fe, respectively [35-39]. Thus, Fe/Fe3O4 is successfully loaded on carbon through an impregnation process followed by annealing.
Figure 1. (a) SEM images, (b) EDS results, (c) TEM, and (d) HRTEM images of AF- 1.12 sample.
Point 3: Figure 3a; The authors claimed that all samples exhibited type â…£ isotherms. However, the hysteresis is not clearly observed in Figure 3a.
Response 3: Thank you for your careful review. We apologize for not accurate describing the adsorption/desorption isotherms of samples. We have modified manuscript based on more related research on page 5, line 4. and Figure 3a.
Figure 3. Ar adsorption/desorption isotherms (a) and corresponding pore size distributions (b) of AC and Fe@C composites.
The surface areas and pore size distributions of AC and Fe@C composites are investigated by Ar adsorption isotherms at 87 K. As shown in Fig. 3(a), all samples display a combined Ι/IV isotherms [15, 35, 37, 42, 49, 52, 53]. Specifically, the rapid rise of isotherms in low-pressure region is a typical character of micro-pores structure [54]. The adsorption/desorption isotherms in high-pressure region exhibit unobvious platform and the type H4 hysteresis loops, indicating the presence of meso-pores [55] The effects of Fe/Fe3O4 on the surface pore size distributions are further studied by QSDFT method. As shown in Fig.3b, all samples have micro-pores (pore size < 2 nm) and meso-pores (pore size between 2 nm and 50 nm), and volume of former is higher than that of latter.
References:
[52] Sun, F.; Qu, Z.; Gao, J.; Wu, H. B.; Liu, F.; Han, R.; Wang, L.; Pei, T.; Zhao, G.; Lu, Y. In situ doping boron atoms into porous carbon nanoparticles with increased oxygen graft enhances both affinity and durability toward electrolyte for greatly improved super-capacitive performance. Advanced Functional Materials 2018, 28, (41), 1804190-1804200.
[53] Li, X. R.; Jiang, Y. H.; Wang, P. Z.; Mo, Y.; Lai, W. D.; Li, Z. J.; Yu, R. J.; Du, Y. T.; Zhang, X. R.; Chen, Y. Effect of the oxygen functional groups of activated carbon on its electrochemical performance for supercapacitors. New Carbon Materials 2020, 35, (3), 232-243.
[54] Li, Z. N.; Gadipelli, S.; Li, H. C.; Howard, C. A.; Brett, D. J. L.; Shearing, P. R.; Guo, Z.; Parkin, I. P.; Li, F. Tuning the interlayer spacing of graphene laminate films for efficient pore utilization towards compact capacitive energy storage. Nature Energy 2020, 5, (2), 160-168.
[55] Peng, Z. Y.; Hu, Y. J.; Wang, J. J.; Liu, S. J.; Li, C. X.; Jiang, Q. L.; Lu, J.; Zeng, X. Q.; Peng, P.; Li, F. F. Fullerene based in situ doping of N and Fe into a 3D cross like hierarchical carbon composite for high performance supercapacitors. Advanced Energy Materials 2019, 9, (11), 1802928.
Point 4: Table S2; The specific surface area of AF samples decreased with increasing Fe contents due to destruction of microporous structure. However, the bare AC showed a smaller specific surface area than that of AF-0.48. What is the reason for the discrepancy?
Response 4: As reviewer suggested, we have given a more detailed explanation about the effects of Fe3O4 loading on the specific area (Page 5, line 11).
Table S3. Specific surface areas of AC and Fe@C composites are investigated by Ar adsorption isotherms at 87 K.
AC |
AF-0.48 |
AF-0.8 |
AF-1.12 |
AF-1.6 |
|
Specific surface area (m2 g-1) |
1503.9 |
1676.81 |
1621.04 |
1518.52 |
1586.46 |
In Table S3, as Fe-doping content rises, the specific area increases from 1503.9 m2 g−1 (AC) to 1676.81 m2 g-1 (AF-1.12) and then decreases to 1586.46 m2 g−1 (AF-1.6). AF-1.2 exhibits the highest specific area with 1676.81 m2 g-1, 11.2 % higher than that of AC. The larger surface area can be attributed to the increase of the microporous volume by etching effect of Fe3O4 during the heat treatment process [56, 57] (Fig. 3b). The specific area of AF-1.6 is 1586.46 m2 g−1, which is smaller than that of AC-1.2. The reduce of specific surface area can be attributed to the reduction of a partial of microporous volume, which is due to the destruction of microporous structure on carbon surface caused by the massive growth of Fe/Fe3O4 [58].
References:
[56] Fan, H. L.; Niu, R. T.; Duan, J. Q.; Liu, W.; Shen, W. Z. Fe3O4@carbon nanosheets for all-solid-state supercapacitor electrodes. ACS Applied Materials & Interfaces 2016, 8, (30), 19475–19483.
[57] Nawwar, M.; Poon, R.; Chen, R.; Sahu, R. P.; Puri, I. K.; Zhitomirsky, I. High areal capacitance of Fe3O4 decorated carbon nanotubes for supercapacitor electrodes. Carbon Energy 2019, 1, (1), 124-133.
Point 5: Figure 4d; The authors claimed that the Rohm values decreased due to the growth of Fe content in the nanocomposites with high electrical conductivity. However, Fe/Fe3O4 composite might have much lower electrical conductivity compared to the carbon. Explain this.
Response 5: As reviewer suggested, we have presented a more detailed statistical analysis on the EIS section (Page 6, line 23).
To gain further insights into the effects of Fe and Fe3O4 on the electrochemical performances, CV and EIS curves are recorded and the results are gathered in Fig. 4c-d. The CV curves of Fe@C composites display asymmetrical pseudo-capacitance behavior (Fig. 4c), corroborating the GCD curves. The asymmetrical CV curve of AF-1.12 suggests the presence of pseudo-capacitance during the charge and discharge processes [41, 59]. In Fig 4d, the X-intercepts of the semicircles in Nyquist plots in the high-frequency region are related to the ohmic resistance of the SCs (Rohm), which represents the resistance of the electrolyte, current collectors and electrodes [28, 29]. Among all the samples, the AC exhibits the highest Rohm of 3.88 Ω, which can be attributed to low charge-transfer rate caused by its uneven surface and the intricate conductive network. While with increasing the loading amount of Fe and Fe3O4, the Rohm values decrease to 1.08, 0.94 and 0.71 Ω for AF-0.48, AF-0.8, AF-1.12 and AF-1.6, respectively. Such significant decrease is the result of that Fe element contributed to structural rearrangement and more active sites on carbon surface, which accelerates charge separation and transport [60-63]. In the low-frequency region, the impedance plots of all samples become oblique lines, showing pure capacitance.
Figure 4. Cycle performances (a), GCD curves (b) at 1 A g−1, CV curves (c) at 1 mV/s, and Nyquist plots (d) of AC and Fe@C composites SCs electrodes.
Reference:
[62] Li, Q. K.; Li, X. F.; Zhang, G.; Jiang, J. Cooperative spin transition of monodispersed FeN3 sites within graphene induced by CO adsorption. Journal of the American Chemical Society 2018, 140, (45), 15149–15152.
[63] Zhao, L.; Zhang, Y.; Huang, L. B.; Liu, X. Z.; Zhang, Q. H.; He, C.; Wu, Z. Y.; Zhang, L. J.; Wu, J. P.; Yang, W. L.; Gu, L.; Hu, J. S.; Wan, L. J. Cascade anchoring strategy for general mass production of high-loading single atomic metal nitrogen catalysts. Nature Communications 2019, 10, (1), 1278.

Reviewer 3 Report
This paper study effects of Fe on self-discharging of C-based supercap. This paper is well written and has interesting content and is very relavant to topic of MDPI materials.
I recommend the publication of this paper after minor modifications.
1) Don't need to include survey scan of XPS. Please delete Figure 2c.
2) original data (grey color) has large variation and noise. Any specific reasons in Figure 2f.
3) specific capacitance has highest value at AF 1.12 not AF 1.6. Any reason?
4) There are some typo. Please look into all sentence in revision carefully.
5) Please add sub Figure like FIgure 6a, 6b, 6c.... in Figure 6.
Author Response
Response to Reviewer 3 Comments
Thanks for reviewers’ comments concerning our manuscript entitled “Effects of Fe Impurities on Self-discharge Performance of Carbon-based Supercapacitors” (Manuscript ID: materials-1163098). These comments are all valuable and very helpful for revising and improving our paper, as well as the important guiding significance to our researches. We have studied comments carefully and have made corrections which we hope meet with approval.
Reviewer 3:
This paper study effects of Fe on self-discharging of C-based supercap. This paper is well written and has interesting content and is very relavant to topic of MDPI materials.
I recommend the publication of this paper after minor modifications.
Point 1: Don't need to include survey scan of XPS. Please delete Figure 2c.
Response 1: Thanks for review’s kind reminder. We have deleted Figure 2c and revised manuscript on page 4, line 20. And, we have changed “Fig. 2d” to “Fig. 2c”, “Fig. 2e” to “Fig. 2d” and “Fig. 2f” to “Fig. 2e”.
Figure 2. (a) XRD patterns and (b) Raman spectra of AC and Fe@C composites. High resolution XPS survey of AF-1.12: (c) C 1s, (d) O 1s, and (e) Fe 2p.
To determine the valence states of Fe and its oxides, XPS measurements are carried out. As shown in Fig. S1, the XPS survey scan unambiguously indicates the presence of C, O, and Fe elements. The raise in Fe/Fe3O4 content leads a gradual enhancement in the peak intensity of Fe 2p corresponding to Fe@C composite when compared to that of AC. The high-resolution spectra of C 1s are divided into six peaks at of 284.7 eV, 285.4 eV, 286.2 eV, 287.4 eV, 288.7 eV and 290 eV, corresponding to sp2 (graphitic carbon, C-C peak), sp3 hybridized carbon (defects peak), C-O, C=O, O-C=O and π-π*, respectively (Fig. 2c) [24, 43, 50, 51]. As shown in Fig. S2 and Table. S2, an increase in Fe-doping content leads to the gradual transformation of sp2-C into sp3-C, indicating the decrease of graphitization degree [51]. This result is consistent with the XRD and Raman results. In Fig. 2d, the O 1s could be deconvoluted into three peaks at 530.2, 531.8 and 533.2 eV, corresponding to O-Fe of Fe3O4, O=C and O-C, respectively [40, 41, 43, 50]. In the Fe 2p spectrum of AF-1.12 (Fig. 2e), five peaks at 710.8, 712.7, 719.0, 724.5 and 725.2 are observed. These are associated with respectively Fe 2p3/2, shake-up satellite and Fe 2p1/2 of Fe2+ and Fe3+, indicating the presence of Fe3O4 [28, 36, 37, 50]. As a result, a unique complex system consisting of Fe@C with different doping concentrations is obtained.
Point 2: original data (grey color) has large variation and noise. Any specific reasons in Figure 2f.
Response 2: Thanks for your careful checks. This may be caused by two factors. On the one hand, the reason may be is that the Fe content of AF-1.12 is too little (6952.0 mg kg-1) to be obtained by X-ray photoelectron spectroscopy (XPS). On the other hand, Fe and Fe3O4 were loaded on porous structure of activated carbon, which also increases the difficulty of XPS measurement.
Table S1. The amounts of Fe in AC and Fe@C composites are investigated by ICP-MS.
AC |
AF-0.48 |
AF-0.8 |
AF-1.12 |
AF-1.6 |
|
Fe (mg kg-1) |
425.0 |
3260.2 |
4625.3 |
6952.0 |
9911.5 |
Point 3: specific capacitance has highest value at AF 1.12 not AF 1.6. Any reason?
Response 3: Thanks for review’s kind reminder. As shown in Fig. 3b and Table. S3, the micro-pores volume of AF-1.2 is larger than AF-1.6, which improved large adsorption space of electrolyte ions, resulting in an increase in double-layer capacitance. And we have given a more detailed explanation about the effects of specific area on the specific capacitance (Page 6, line 2) as follow:
The capacitive performances of AC and Fe@C composites are investigated in 6 M KOH for over 10,000 cycles. As shown in Fig. 4a, as Fe-doping content rises, the specific capacitance increases from 146.7 F g−1 (AC) to 186.2 F g−1 (AF-1.12) and then decreases to 173.2 F g−1 (AF-1.6). AF-1.2 exhibits the highest specific capacitance with 186.2 F g-1, 26.9 % higher than that of AC. The enhanced electrochemical performance can be attributed to the introduction of Fe/Fe3O4, which increases the pseudocapacitance by redox reactions in the interface between electrode and electrolyte. The specific capacitance of AF-1.6 is 173.2 F g−1, which is smaller than that of AC-1.2. The reduce of specific capacitance can be attributed to the reduction of a partial of double-layer capacitance caused by the destruction of microporous structure and the reduce of specific surface areas. After 10,000 cycles, the specific capacitance of AF-1.12 remains at 178 F g−1, indicating excellent cycle stability (Figure 4a). The degradation in long-cycle performance is caused by the instability of increased pseudocapacitance by Fe and its oxide. The GCD curves of Fe@C composites look nonlinear and asymmetrical, suggesting a strong pseudocapacitive behavior (Fig 4b) [44, 51, 58].
Figure 3. Ar adsorption/desorption isotherms (a) and corresponding pore size distributions (b) of AC and Fe@C compo-sites.
Table S3. Specific surface areas of AC and Fe/Fe3O4 composites are investigated by Ar adsorption isotherms at 87 K.
AC |
AF-0.48 |
AF-0.8 |
AF-1.12 |
AF-1.6 |
|
Specific surface area (m2 g-1) |
1503.9 |
1676.81 |
1621.04 |
1518.52 |
1586.46 |
Figure 4. Cycle performances (a), GCD curves (b) at 1 A g−1, CV curves (c) at 1 mV/s, and Nyquist plots (d) of AC and Fe@C composites SCs electrodes.
Point 4: There are some typo. Please look into all sentence in revision carefully.
Response 4: We apologize for the language problems in the original manuscript. The language presentation was improved with assistance from a native English speaker with appropriate research background.
Point 5: Please add sub Figure like Figure 6a, 6b, 6c.... in Figure 6.
Response 5: Thanks for review’s kind reminder. We have changed Figure 6 and the related to the revised manuscript on Page 7, line 5, Page 7, line 13.
Figure 6. (a) Illustration of the experiment, The mechanisms of SDC in SCs: (b) ohmic leakage mechanism, (c) Faraday reaction mechanism, and (d) diffusion mechanism.
The main form of self-discharge of charged SCs relies on voltage attenuation [12, 64]. Hence, the devices are charge/discharge two and a half cycles (at 1 A g-1) and then hold at 1 V for 2 h to reduce the influence of uneven charge on the electrode surface. As shown in Fig. 5a, the potential of AC drops to 0.27 V after 24 h, equivalent to 73% voltage loss in the SCs. By comparison, AF-0.48 takes the same time to drop its potential to 0.46 V, equivalent to only 54% voltage loss. The increase of Fe and Fe3O4 contents leads to the gradual decline in voltage loss of SDC. Compared with the potential decay of AC, the that of AF-1.12 reduces by 39.7% (0.56 V, equivalent to only 44% voltage loss) after 24 h. According to the mathematical models proposed by Conway et al., the potential decay is caused by three mechanisms of SCs consisting of ohmic leakage, Faraday reaction, and diffusion (Fig. 6) [12, 14]. The potential decay is caused by the internal resistances between the positive and negative electrodes, which can be described by the ohmic leakage mechanism in Fig. 6b (V ~ exp(t)) [64, 65]. As shown in Fig. 6c, the potential decay is attributed to the departure of ions from the electric double layer during SDC process, which is the diffusion mechanism (V ~ t1/2) [12, 24]. The potential decay is related to the redox reactions of impurities on the electrode surface, which can be described by the Faradic redox mechanism (V ~ ln (t)) (Fig. 6d) [14, 24, 63]. Combining the potential decay causes by all three mechanisms, Eq. (1) could be obtained [64, 66].
(1) |
|
( |
where U and U0 represent the final potential and initial potential (1.0 V) of the charged SC. t is the discharge time (24 h). A and B correspond to the time constant and diffusion parameter of the SDC process, respectively. e and D denote the constants in the Faradaic process, related to the current density.

Round 2
Reviewer 1 Report
The authors answered to the Reviewer's observations and improved the manuscript.
Author Response
Response to Reviewer 1 Comments
Thanks for reviewers’ comments concerning our manuscript entitled “Effects of Fe Impurities on Self-discharge Performance of Carbon-based Supercapacitors” (Manuscript ID: materials-1163098). These comments are all valuable and very helpful for revising and improving our paper, as well as the important guiding significance to our researches. We have studied comments carefully and have made corrections which we hope meet with approval.
Reviewer 1:
Point 1: The authors answered to the Reviewer's observations and improved the manuscript.
Response 1: Thanks for review’s recognition. In addition, we have carefully checked grammar and spelling of the article. The language presentation was improved with assistance from a native English speaker with appropriate research background, which we hope meet with approval.

Reviewer 2 Report
I think the authors revised the manuscript properly. Thus I support the acceptance of this manuscript.
1. Figure 1c inset; The author claimed that the average size of Fe or Fe3O4 particles is about 0.45 nm. Isn't it 4.5 nm? Please, double check the particle size.
Author Response
Response to Reviewer 2 Comments
Thanks for reviewers’ comments concerning our manuscript entitled “Effects of Fe Impurities on Self-discharge Performance of Carbon-based Supercapacitors” (Manuscript ID: materials-1163098). These comments are all valuable and very helpful for revising and improving our paper, as well as the important guiding significance to our researches. We have studied comments carefully and have made corrections which we hope meet with approval.
Reviewer 2: I think the authors revised the manuscript properly. Thus I support the acceptance of this manuscript.
Point 1: Figure 1c inset; The author claimed that the average size of Fe or Fe3O4 particles is about 0.45 nm. Isn't it 4.5 nm? Please, double check the particle size.
Response 1: Thanks for review’s kind reminder. As reviewer suggested, we have carefully checked the average size of Fe or Fe3O4 particles and modified the related section of Figure 1c on revised manuscript as follow (Page 3, line 19):
Figure 1. (a) SEM images, (b) EDS results, (c) TEM, and (d) HRTEM images of AF- 1.12 sample.
The surface morphologies of Fe@C composites are investigated by SEM and the results are displayed in Fig. 1a. The annealed Fe@C composites present irregular particles with porous structure. The porosity between the disordered particles and porous structure would provide favorable space for the penetration and diffusion of the electrolyte, useful for improving the capacitance of the SCs [28-30]. Besides, the Fe, C, and O elements are also uniformly distributed on the particle surface (Fig. 1b). The crystal structures of Fe@C composites are further investigated by TEM and HRTEM measurements. As shown in Fig. 1c and 1d, Fe and its oxide are uniformly anchored on the activated carbon surface, and the average size of particles in AF-1.12 is about 4.5 nm (inset picture of Figure 1c). Meanwhile, the porous structure of activated carbon particles could provide enough space for the multiphase changes of Fe3O4 and prevent huge volumetric expansion during the charge-discharge processes to maintain high cycling stability [31-34]. More direct evidence regarding the structure of Fe@C composites is illustrated in Fig. 1d. For AF-1.12 sample, the lattice distances are estimated to 0.253 nm and 0.202 nm attributed to (331) lattice planes of Fe3O4 and (110) lattice planes of Fe, respectively [35-39]. Thus, Fe/Fe3O4 is successfully loaded on carbon through an impregnation process followed by annealing.
